# Nanotechnology as a Promising Approach to Combat Multidrug Resistant Bacteria: A Comprehensive Review and Future Perspectives

**DOI:** 10.3390/biomedicines11020413

**Published:** 2023-01-31

**Authors:** Helal F. Hetta, Yasmin N. Ramadan, Alhanouf I. Al-Harbi, Esraa A. Ahmed, Basem Battah, Noura H. Abd Ellah, Stefania Zanetti, Matthew Gavino Donadu

**Affiliations:** 1Department of Medical Microbiology and Immunology, Faculty of Medicine, Assiut University, Assiut 71515, Egypt; 2Department of Microbiology and Immunology, Faculty of Pharmacy, Assiut University, Assiut 71515, Egypt; 3Department of Medical Laboratory, College of Applied Medical Sciences, Taibah University, Yanbu 46411, Saudi Arabia; 4Department of Pharmacology, Faculty of Medicine, Assiut University, Assiut 71515, Egypt; 5Department of Biochemistry and Microbiology, Faculty of Pharmacy, Syrian Private University (SPU), Daraa International Highway, 36822 Damascus, Syria; 6Department of Pharmaceutics, Faculty of Pharmacy, Assiut University, Assiut 71515, Egypt; 7Department of Pharmaceutics, Faculty of Pharmacy, Badr University in Assiut, Naser City, Assiut 2014101, Egypt; 8Department of Biomedical Sciences, University of Sassari, 07100 Sassari, Italy; 9Hospital Pharmacy, Azienda Ospedaliero Universitaria di Sassari, 07100 Sassari, Italy

**Keywords:** nanotechnology, nanoantibiotics, MDR bacteria, XDR bacteria, biofilm

## Abstract

The wide spread of antibiotic resistance has been alarming in recent years and poses a serious global hazard to public health as it leads to millions of deaths all over the world. The wide spread of resistance and sharing resistance genes between different types of bacteria led to emergence of multidrug resistant (MDR) microorganisms. This problem is exacerbated when microorganisms create biofilms, which can boost bacterial resistance by up to 1000-fold and increase the emergence of MDR infections. The absence of novel and potent antimicrobial compounds is linked to the rise of multidrug resistance. This has sparked international efforts to develop new and improved antimicrobial agents as well as innovative and efficient techniques for antibiotic administration and targeting. There is an evolution in nanotechnology in recent years in treatment and prevention of the biofilm formation and MDR infection. The development of nanomaterial-based therapeutics, which could overcome current pathways linked to acquired drug resistance, is a hopeful strategy for treating difficult-to-treat bacterial infections. Additionally, nanoparticles’ distinct size and physical characteristics enable them to target biofilms and treat resistant pathogens. This review highlights the current advances in nanotechnology to combat MDR and biofilm infection. In addition, it provides insight on development and mechanisms of antibiotic resistance, spread of MDR and XDR infection, and development of nanoparticles and mechanisms of their antibacterial activity. Moreover, this review considers the difference between free antibiotics and nanoantibiotics, and the synergistic effect of nanoantibiotics to combat planktonic bacteria, intracellular bacteria and biofilm. Finally, we will discuss the strength and limitations of the application of nanotechnology against bacterial infection and future perspectives.

## 1. Introduction

Antimicrobial resistance poses a serious threat to global public health, causing at least 1.27 million deaths globally and approximately 5 million fatalities in 2019. Each year, more than 2.8 million illnesses in the US are resistant to antibiotics. More than 35,000 people die as a result, according to CDC’s 2019 Antibiotic Resistance (AR) Threats Report [1].

Antimicrobial resistance may have an impact on individuals at every stage of life, as well as on the medical, veterinary, and agricultural sectors [2,3,4,5,6,7]. This makes it one of the most important public health issues in the entire world [1].

Rapid development of Ab resistance in microorganisms is directly related to overconsumption of Abs, their broad utilization in agriculture and the non-availability of novel antibiotics [8,9]. Due to the fact that repeated drug administration and greater doses are common nowadays, Antibiotic resistance has emerged against various types of antibiotics commonly used against harmful bacteria [10]. Development of Antibiotic resistance has a severe impact on the efficacy of antimicrobial medication and is linked to high fatality rates and high medical costs [11].

To be hazardous, bacteria do not necessarily need to be resistant to all antibiotics. Serious issues can result from resistance to even a single antibiotic. For example, second- and third-line treatments for antibiotic-resistant infections can harm patients by resulting in severe adverse effects, such as organ failure, and delaying treatment and recovery for weeks or even months. If antibiotics and antifungals become ineffective, we will be unable to treat infections and manage these public health problems [1,12,13,14,15,16]. 

To overcome this serious problem, nanotechnology and nanoparticles were developed in recent years to reduce and combat bacterial resistance, multidrug resistance (MDR) and even bacterial biofilm. Nanoparticles are engineered structures with sizes ranging from 1 to 100 nm (nm) [17]. Nanomaterials are used in a variety of medical applications, ranging from medical equipment to therapeutic agents, drug delivery systems, and diagnostic imaging systems [17].

Nanotechnology-based delivery systems are increasingly being evaluated as viable options for improving therapeutic efficacy by limiting drug degradation, increasing accumulation at infection sites, and reducing toxicity [18,19,20,21]. The NPs provide the most effective method of addressing MDR bacteria since they not only act as transporters for natural antibiotics and antimicrobials, but also actively combat bacteria. Moreover, loaded NPs have the capacity to safely and effectively transport a wide variety of treatments to the site of infection when they are bonded to their surface or enclosed within a structure [11,22,23,24,25].

Nanoparticles can act as inherent therapies (e.g., titanium dioxide (TiO2), zinc oxide (Zn O) NPs [26], copper (Cu) NPs [27], nickel (Ni) NPs) [28], selenium (Se) NPs [28,29,30], silver (Ag) NPs) [31,32] or as nano-carriers for antimicrobial agents “Nanobiotics” (e.g., liposomes, polymeric NPs, and dendrimers) [33,34].

Due to the ability of the NPs to carry medications to the target location in the ideal dosage range, protect them against deactivation, and boost their therapeutic effectiveness with fewer adverse effects, nanoparticles are considered a targeted therapy approach [22,24,35,36,37,38,39,40,41,42].

The nano-formulations’ small size, large surface area, and highly reactive nature allow them to penetrate biological barriers such as biofilm and have a high selectivity for bacterial cells [23,43,44]. The increased surface area of NPs makes it easier to load medications while the NPs sizes are tiny enough to penetrate biofilms and microbial cell walls. Additionally, NPs are quickly excreted through the kidney and have extended plasma half-lives [43,45].

Typically, the synthesis procedures and formulation conditions of nanoparticles, such as the type of reducing agent, temperature, concentration, and solvent, typically determine their antibacterial or biological properties [46,47].The behavior and activity of nanoparticle are significantly determined by chemical composition, shape, size and size distribution [48]. The antimicrobial properties of nanoparticles are also influenced by a number of factors, including temperature, pH, metal ion concentration, cell age, and reaction time [49].

Organic nanoparticles (e.g., liposomes, polymeric nanoparticles, micelles, ferritin) and inorganic nanoparticles (e.g., metal nanoparticles) have both been used to treat a variety of medical diseases [50]. Drug transport efficiency, bioavailability, and antibacterial activity have all been improved with the use of organic nanoparticles [51]. They have showed promise against viral infections, such as severe acute respiratory syndrome 2 (SARS-CoV-2) and have also been used to treat fungal infection [52,53,54,55,56].

The present variety of nanoparticulate antibacterial systems consists of liposomes, micelles, solid lipid nanoparticles, carriers nanogels, nanocapsules, nanotubes, dendrimers, emulsions, nanostructured lipid, quantum dots and polymeric NPs. They are slow-release nanoscale drug delivery systems that deliver drugs to targeted cells [54,57].

The most promising ones seem to be the metallic NPs. Several multi-drug resistant microorganisms are responsive to a variety of their activities [53,58]. Among all the nanoparticles, AgNps is the most studied and utilized [59].It is crucial to characterize metal NPs to understand their toxicity and mode of action. A variety of methods including UV–visible spectroscopy, single-particle inductively coupled plasma mass spectrometry, X-ray diffraction analysis, transmission electron microscopy (TEM), atomic force microscopy, Zeta potential measurement, dynamic light scattering and Fourier transform infrared spectroscopy analysis can be used for this purpose [60,61].

In this review, we will discuss the ways in which the bacteria develop resistance to conventional antibiotics, serious widespread of MDR and XDR bacteria. Moreover, we will discuss NPs as a trend approach to combat MDR organisms, their types and formulation and expected mechanism of action. Additionally, we will explain synergetic effect of NPs loaded with conventional antibiotics and the way this affects targeting of antibiotic and potentiate bacterial cell lysis as shown in Figure 1.

## 2. Development of Antibiotic Resistance 

Bacteria are living entities that change over time. Their major goal is to multiply, endure, and spread as quickly as they can. Microbes, as a result, adapt to their environment and develop in ways that ensure their survival [62,63]. If anything, such as an antibiotic, inhibits their ability to develop, genetic changes may occur, rendering the bacteria resistant to the medicine and enabling them to survive [64].

Antibiotics can be rendered inactive by bacteria via a variety of molecular mechanisms [65] (Figure 2).

1.Production of inactivating enzymes: The bacterium produces particular enzymes that precisely inactivate the antibiotic, causing it to lose its biological function. For instance, when β-lactam drugs are digested by β-lactamases, this occurs. Extended-spectrum β-lactamases (ESBLs), which have the same inactivating activity, are produced by some bacteria, making them challenging to eradicate. The enzymes acetyltransferase, phosphotransferase, and adenyl transferase are additional ones that can render certain antibiotics inactive [66,67,68,69].2.Changes and alterations in the antibiotic target: In the case of erythromycin resistance, for instance, methylation of an adenine residue in the peptidyl-transferase of r-RNA 23S reduces the enzyme’s affinity for the antibiotic without impairing protein synthesis. The alteration of penicillin-binding proteins (PBPs) by MRSA is another significant instance [70].3.Decrease in antibiotic uptake: Changes in the structural architecture of the cell’s surface casings can reduce the entry of an antibiotic. For example, a change or reduction in the quantity of porins or porin gene mutation may be the cause of the resistance in Gram-negative bacteria [71,72,73].4.Extrusion of drug outside the cell: Energy-driven drug efflux systems eliminate antibiotics that have been taken up by bacterium cells. Many times, bacterial cells that are resistant to antibiotics have upregulated efflux pumps [71,73,74,75,76].5.Activation of alternative metabolic pathways: The sulfonamides case serves as an explanation. In fact, bacteria exposed to sulfonamides are still able to produce folic acid through a different metabolic pathway [77].6.Transferring of resistance genes: As a result of bacterial communication and genetic information sharing, resistance genes propagate among bacterial populations [8,78].7.Inactive metabolic state of microorganism: The metabolically inert bacterial subpopulations known as persister cells have a lower antibacterial susceptibility than active ones [79].8.Extracellular polymeric substances (biofilm) formation: Due to electrostatic repulsion, some antibiotics, such aminoglycosides, are unable to penetrate the protective extracellular polymeric components of bacterial biofilms [78,80,81].

## 3. Spread of MDR and XDR Bacteria

Once a species gains resistance to an antibiotic, it eventually overcomes the combined drug effects, leading to development of multidrug-resistant (MDR) organisms which are resistant to antimicrobial drugs or combination treatments. By encouraging bacterial growth that was not immediately eliminated, this scenario occasionally encourages the spread of resistance [82,83,84,85].

Increased antibiotic resistance is among the world’s top three public health issues of the 21st century, according to the World Health Organization (WHO). The much more harmful bacteria, collectively known as ESKAPE *(Enterococcus, Staphylococcus, Klebsiella, Actinobacter, Pseudomonas, and Enterobacter),* are those that cause the most concern. They are all are connected to a high death rate [72,86,87,88]. The spread of MDR and extensively drug-resistant (XDR) bacteria continues to represent a severe threat to human and animal health due to their high mortality, morbidity, and prices of medication to combat them [89,90,91,92]. This is made worse by a number of bacterial strains’ capacity to form biofilms, which is linked to 65–80% of human illnesses [93]. Antibiotic susceptibility of biofilm cells is typically 100–1000 times lower than that of planktonic cells [94]. Drug resistance in biofilms is caused by various factors, including decreased drug absorption throughout the extracellular polymeric matrix, decreased intracellular drug level, decreased bacterial metabolism, and transferring resistant genes [95,96]. MDR causes blockage in the fight against disease and decreases treatment effectiveness, which causes the patient’s infection to last longer than expected. Immunocompromised patients such as those with chronic diseases, receiving cancer chemotherapy, immunosuppressive drugs or undergo organ transplantation are more vulnerable to MDR infection. Additionally, the treatment cost has also increased as a result of MDR since microorganisms have developed resistant to commercially accessible Abs, necessitating the adoption of a more expensive one [97] (Figure 3). The potential for MDR to spread over the world is boosted by the expansion of international trade and travel.

According to reports, multidrug-resistant bacteria are responsible for at least 700,000 deaths annually across the globe, including 23,000 deaths in the US and 25,000 deaths in the EU [98]. According to reports from the WHO, misuse and overuse of antibiotics account for approximately 80% of MDR or XDR bacteria, and these infections are linked to serious negative effects [87]. It is also predicted that if no action is taken to combat bacterial resistance or create new medicines, 10 million individuals will die globally from bacterial diseases by the year 2050 [99]. As MDR pathogens and other resistant organisms proliferate, there are currently few therapeutic and preventative options available [87]. The lack of success with conventional antibiotics forces the quest for more effective treatments. Most bacterial infections can be successfully treated using nanoparticles, particularly those caused by MDR pathogens. Antibiotics and nanoparticles can be employed separately or in combination to produce strong synergistic effects [54].

## 4. Types and Formulations of Nanoparticles

### 4.1. Organic NPs

Organic NPs have been extensively studied over the recent years and contain a variety of types. The first nano-scale medication licensed for medical use is called a liposome [100]. Liposomes are biocompatible, biodegradable, and nontoxic vesicles. They consist of one or more outer lipid layers and core holding hydrophilic or hydrophobic medication [101,102]. 

Liposomal Np core has high drug loading capacity, and so, by encapsulating hydrophilic antimicrobial drugs in liposomal NP core, it is possible to protect them from degradation in vivo. Bassetti M et al. studied the effect of aerosolized liposomal antimicrobials (ciprofloxacin, tobramycin, amphotericin B and amikacin) and their curative significant effect to prevent and treat acute and chronic respiratory tract infection [103].

Another type of organic NPs is polymer-based, which has unique structural arrangements for drug delivery created by numerous monomers [104,105]. Polymeric NPs include polylactic-co-glycolic acid (PLGA), which is formed through co-polymerization of lactic and glycolic acid [106,107]; dendrimers, which are characterized with a 3D branching architecture [108]; and micelles, which are distinguished by polymer self-assembly into nano-aggregates [109].

### 4.2. Inorganic NPs

Inorganic NPs may be categorized into metal nanoparticles and nanoparticles made of metal oxide. They are characterized with their own antibacterial activities. Due to their unique antibacterial activities, they may be used alone or loaded with antibiotic and exert dual mechanisms (Figure 4) [110].

Metal NPs my include Ag [110,111], Au [112,113], Cu [27,110], Si [114], Ni [28], and Se [28,30], and metal oxide may include ZnO [26,115,116], TiO_2_ [117,118,119], CuO [120], MgO [116] and SiO_2_ [121].

Organic NPs have advantage over inorganic type as they have higher surface area to volume ratio. Unfortunately, they have poorer biodegradability and biocompatibility [122].

Inorganic NPs may have some sort of cytotoxicity and this effect depends on their size and charge. This cytotoxicity is dose-related [123].

Carbon nanotubes are tubes made of carbon with diameters typically measured in nanometers, characterized by a vast promise in the drug delivery area due to its distinct biological, physical, and chemical properties [124,125].

Mesoporous nanoparticle carriers, which are subtypes from silica NPs, have high surface area, strong framework, and are porous. This porous nature creates high content of internal pores and so is loaded with maximum number of antimicrobials [126,127,128]. 

### 4.3. Hybrid NPs

Hybrid NPs are formed from combination of organic and inorganic NPs in a single composite drug delivery system. Therefore, this composite system improves biological features and increases treatment efficacy and decreases toxicity and resistance [129].

Lipid polymer hybrid NPs are constructed from polymeric core and outer lipid sheath. It has been established and proved that these hybrid NPs are an effective drug delivery system [130]. These hybrid NPs combine the advantages of both lipids and polymers; therefore, they are characterized with biocompatibility, structural rigidity, and sustained drug release. This type can also be loaded with both hydrophilic and hydrophobic drugs and achieve perfect drug targeting and release [131,132].

In recent study, it was shown that the hybridization may include polymer gold (AuNP). In the study, He X et al. examine the effect of polyhexamethylene biguanide polymer hybridized with AuNP and prove a promising synergistic activity of this drug delivery system to propagate wound healing infected with *S. aureus* [133]. 

Moreover, magnetic carboxymethyl cellulose-ε-poly-lysine hybrids were synthesized and used as matrix to anchor Ag NPs. This hybrid Ag NP complex shows promising antibacterial activity against *S. aureus* and *E. coli* and shows synergistic antibacterial activity for wound infection treatment [134]. 

## 5. Free Antibiotic VS Nanoantibiotic

Antibiotics are a substance that reduces the growth of germs, combats some diseases, and saves lives when administered properly. They are the basic pharmacological options for treating biofilm and planktonic infections [135,136].

Nonoantibiotic (nAbts) are pure antibiotic molecules that have been synthetically grown to an average diameter of ≤100 nm minimally at one dimension, or engineered NPs that encapsulate the antibiotic molecules.

Antibiotics frequently result in the death of bacteria by disrupting the proton driving force throughout the cell membrane, lowering the microbe ability to produce or store energy, blocking protein production, rupturing the structural components of the cell wall, etc. [137].

Multifunctional “intelligent” antimicrobials contained in NPs react to stimuli to interact with the surfaces of bacterial cell walls and membranes, improving entry through membranes and drug distribution to target sites. nAbts further have a target-specific, controlled release which can be administered in a single dose, unlike the majority of traditional antibiotics that require many doses in a systematic release [138].

In comparison to “free” antibiotics, the dosage of antibiotics administered via nanoparticle-based delivery systems has additional benefits, including decreased amount and frequency of the dose, sustained drug release that boosts drug efficiency, and targeted drug delivery to a specific site within bacterial cell [138,139].

When compared to molecular materials, NPs have a high surface-to-volume ratio and a unique surface area; as a result, these nanoscale particles have an extraordinary large contact area [140].

When antibiotics are combined with NPs, the antibiotic is transported to the desired locations and the NPs simultaneously activate a variety of antibacterial defenses. Therefore, NPs provide more functionalities compared to the effects of one medicine or the combination of various therapies for treating MDR strains [140].

Taking β-lactam antibiotics and β-lactam nAbts as an example, β-lactam antibiotics destroy microorganisms via numerous approaches, including (a) inhibiting the transpeptidase enzyme responsible for cross linkage of peptidoglycan in bacterial cell wall from functioning, (b) blocking the penicillin-binding proteins (PBPs), the enzymes responsible for synthesis of peptidoglycan in bacterial cell wall, and (c) destroying Ftsl (PBP3), an important protein complex required by living microorganisms [141].

The production of ROS by nAbts has been linked to DNA deterioration, protein denaturation, membrane malfunction, and deterioration of membrane stability [142]. 

The NPs linked to antibiotics can physically damage cellular structures by dissolving and liberating metal ions, resulting in biomechanical damage [143]. Furthermore, signal transmission inhibits ATPase activity and ribosomal subunit binding to tRNA; modifying metabolic activities include protein regulation, metabolism of lipids, carbs, and energy, as well as prevention of bacterial growth, leading to cell death [144,145,146,147]. The interaction of nanosized antibiotics with intracellular component of bacteria is critical in applications such as antimicrobial transportation, drug transporters, cell membrane infiltration to specific locations, and interruption of protein synthesis. As a result, nAbts are seen as a possible alternative to current antimicrobial resistance and treatment methods in clinical infections [148,149].

## 6. Mechanisms of Action of Nanoparticles (NPs)

It is challenging to pinpoint the specific mechanisms of action of NPs because the action is multifactorial in nature [150,151].

To battle bacteria, nanomaterial can use a variety of bactericidal methods, such as the production of reactive oxygen species (ROS), destruction of cell walls and membranes, the distribution of medicines via membrane fusion, and interacting with intracellular elements (e.g., DNA and ribosomes) (Figure 4). It is noted that nanomaterials’ special physico-chemical characteristics, particularly their chelation with bacterial cells, are determined through Van der Waals forces, receptor–ligand interactions, hydrophobic interactions, and electrostatic attractions [152].

### 6.1. Destruction of Cell Wall and Cell Membrane 

The cell membrane of microorganisms has developed to act as a physical barrier to antimicrobials. Teichoic acids, which are found in Gram-positive bacteria’s cell walls, and lipopolysaccharide, which is present in Gram-negative bacteria’s outer membrane, both include phosphate or carboxyl groups that make the surfaces of the bacteria negatively charged. This intensely polar environment hinders hydrophobic antimicrobials’ ability to penetrate membranes, impairing their ability to combat microorganisms [152].

Through electrostatic adsorption to the cell wall, membrane depolarization, decrease in membrane permeability, and loss of membrane fluidity occur, resulting in disruption of energy transmission and cell death [9].

Besides that, a buildup of NPs causes “pits” to form in the bacterial cell wall. As a result, they can enter cells, altering their cell membranes and resulting in structural damage and cell death [153]. By interacting with the bacterial surfaces’ negative charges (through carboxyl or phosphate groups), positively charged ions produced by NPs have a greater bactericidal effect [154,155]. However, because Gram-positive bacteria have a thick coat of peptidoglycan, it is more difficult for nanoparticles (NPs) to enter the bacteria. As a result, NPs only interact with the bacterial surface [156].

### 6.2. Production of Reactive Oxygen Species (ROS)

ROS are byproduct results from oxidative metabolism. They influence cell development, signaling, survival, and death [146]. In addition, they have strong positive redox potential [157].

ROS include hydroxyl radicals (˙OH), superoxide radicals (O^2−^), singlet oxygen (O2), and hydrogen peroxide (H2O2). Different NPs generate various mixtures of ROS, which have different antibacterial characteristics. For instance, MgO NPs and ZnO NPs only produce the (O^2−^) radicals and a mixture of (H2O2) and the (˙OH) radicals, respectively, while Ag and Cu NPs generate all forms of ROS [154].

ROS are brought on by NPs or by disruption of the respiratory chain [157]. Usually, they are kept in low level by scavengers’ molecules as reduced glutathione [158]. 

In a typical environment, ROS generation and removal are proportional. However, under extreme stress, there is too much ROS generation, which leads to alteration in the cell membrane fluidity and results in bacterial damage [155]. Along with oxidative stress, ROS can disrupt macromolecules in the cell, resulting in lipid peroxidation, protein modification, enzyme inhibition, inhibition of electron transport chain and RNA or DNA damage (Figure 2).

It was reported that significant antimicrobial effects caused by production of ROS can be observed in the case of Ag [110,159,160], Cu [110,161], ZnO [115,162], TiO2 [118,163], and iron [164,165,166] NPs.

### 6.3. Binding to and Damaging Intracellular Component

Bacteria’s ability to function and survive depends on their cellular homoeostasis and intracellular signaling pathways. Cell death can result from the engineering of nanomaterials to obstruct these pathways. Changes in protein synthesis, DNA damage, and altered gene expression are some of these disruptions [167,168]. As an illustration, to create pyrimidine-capped AuNPs, 4,6-diamino-2-pyrimidinethiol (DAPT), an analogue of 2-pyrimidinethiol (found in *E. coli*), was added to AuNPs (Au-DAPT). MDR strains of Pseudomonas aeruginosa and E. coli were prevented from growing by these nanoparticles [144].

### 6.4. Destruction of Biofilm Architecture

NPs can destruct biofilm due to their high penetration power. They penetrate dense layers of biofilm (extracellular polysaccharide matrix), and once they evade biofilm, they interact with bacteria and exert bactericidal effect. Charge in NPs is greatly affecting the interaction and penetration of biofilm. For instance, positively charged or cationic NPs are firmly interacting with anionic matrix [169,170]. 

## 7. Synergistic Activity of Nanoantibiotics

When nanoparticles are coated or coupled with other substances, their activity can be significantly improved (Figure 5). In fact, utilizing nanoparticles in conjunction with antibiotics can help to decrease bacterial resistance. Antibiotics may be incorporated in nanoparticles or bonded to their surfaces, protecting Abs from chemicals and enzymes that would break them down. This protection may improve a drug’s therapeutic effectiveness. lowering dosage is required to improve treatment outcomes and reduce host hazard [171,172,173].

The use of nanocarriers can reduce resistance selection by delivering treatments that stimulate numerous modes of action and by targeting cargo release, preventing bacteria from being exposed to sub-minimal inhibitory doses of the drug [174]; In addition, when they serve as antibiotic carriers, penetration of bacterial cell walls becomes easier. The antibiotic then causes damage to the cell wall, allowing the nanoparticles and their complex to enter the body more easily [54].

In previous studies, when amoxicillin was coupled with AgNps, a noticeable drop in bacterial growth was observed [175], AgNps-ampicillin complex displays more antibacterial action than a single amoxicillin dose against *S. aureus, E. coli, and K. mobilis* [176], Enhanced antimicrobial action against *P. aeruginosa* was seen in vitro and in vivo when the gentamicin was loaded into poly(lactide-co-glycolide) (PLGA) nanoparticles [177]. When ampicillin was bonded to the interface of AuNPs and AgNPs, broad-spectrum bactericidal agents were created that bypass the resistance pathways of MDR strains of *P. aeruginosa, Enterobacter aerogenes, and MRSA* [171]. Increasing colistin activity against colistin-resistant *E. coli* by combining it with alginate nanoparticles and small molecules such as components of essential Oils, polyamines, lactic Acid [178], bimetallic nanoparticles (gold/silver), and core shell linezolid (Au@Ag@Lz) complexes shows good bactericidal action against all investigated pathogens, including *MRSA* [179]. 

### 7.1. Effect on Planktonic Bacteria 

Planktonic bacteria are free-living bacteria [180]. Serious infectious conditions such as sepsis and keratitis are frequently caused by planktonic bacteria. The most serious group of planktonic bacteria are collectively called ESKAPE pathogens *(Enterococcus, Staphylococcus, Klebsiella, Actinobacter, Pseudomonas, and Enterobacter)*, and they account for the majority of hospital-acquired infections, complicating the conditions of patients who are often immunocompromised [1,181,182].

Because these microorganisms produce resistance quickly even to the latest antibiotics, there are few treatment options for infections caused by them [182].

Because little to no resistance development is shown with nanomaterial-based techniques, nanoparticles can save therapeutic design in this regard [9,183,184].

Protein is the primary driver of bacterial life processes and is found in essential bacterial cell components such as cell walls, cell membranes, ribosomes, nucleic acid, etc. As a result, any of the above components can be destroyed and cause bacterial death [185].

The effectiveness of nanomaterials against the ESKAPE pathogens has been the subject of numerous studies: through disrupting cell membrane [111,117,186,187,188], interfering with protein [113,189,190,191,192], or by targeting nucleic acid [193,194,195]. 

### 7.2. Effect on Intracellular Bacteria

Bacteria can live inside of mammalian cells, resulting in recurrent systemic infections. For instance, *Salmonella enterica* is a classic facultative intracellular organism that annually causes potentially fatal food-borne infections in millions of people. Salmonella species have the ability to live and multiply within host cells, including macrophages [196,197]. Because many antibiotics are unable to cross the membranes of mammals and can actively be transported outside by the host cell, intracellular localization of bacteria makes treatment more challenging. Owing to nanomaterials’ capacity to enter eukaryotic cells and their high drug loading capacity, they can help to overcome this obstacle [198,199].

Another example of an intracellular microorganism which persists inside macrophages is *Mycobacterium tuberculosis* (*MTb*) and *Mycobacterium avium* [200,201,202,203]. The efficacy of nanoparticles against intracellular *MTb* species has been shown in multiple studies on topics such as the following: (a) biodegradable multimetallic microparticles including Ag NPs and ZnO NPs were created for pulmonary medication delivery to the endosomal pathway of *MTb*-infected macrophages [204]; (b) ZnONPs and the mixture MgONPs-ZnONPs have higher bactericide behavior and might have synergistic effects against MDR-*MTb* [116]; (c) the use of all trans retinoic acid-loaded nanoparticles in a host-directed therapy for TB, which targets the host’s immune system response rather than *MTb* directly, has promise for enhancing current treatments and reducing the emergence of MDR-*MTb* [107]; (d) PLGA NP loaded with various antibiotics (levofloxacin, linezolid, ethambutol, prothionamide, and pyrazinamide) show promising approach for MDR-*MTb* by enhancing drug efficacy and inducing innate bactericidal events in macrophages [205]. Other studies have proven the promising effect of NPs on other intracellular bacteria: (a) docosanoic acid solid lipid nanoparticles that were loaded with enrofloxacin raised the intracellular concentration of the drug by up to 40 times and improved killing of *Salmonella* within macrophages [206]; (b) colistin, a weakly diffusible antimicrobial agent, was transformed into liposomes complexed with a protein derived from bacteria to promote internalization into eukaryotic cells and deliver drugs with a high bioavailability [207]; (c) *Listeria monocytogenes* and *P. aeruginosa* were eliminated from inside invaded macrophages by gentamicin-loaded AuNPs coated with phosphatidylcholine [208].

### 7.3. Effect on Bacterial Biofilm

A biofilm is a network of thousands of bacteria that acts as a barrier against the human immune system, extreme conditions, and antibiotics [209].

Bacteria can grow in two different ways: as free-floating planktonic organisms or as sessile, surface-attached communities seen in biofilms, which are organized colonies enclosed in a self-produced extracellular polymeric matrix (EPS) [210,211].

Biofilm formation is a process initiated by planktonic (free-living) organisms aggregating on the outside of either living or inanimate items [212].

Approximately two-thirds of the biofilm’s volume is composed of EPS [213], which mostly consists of lipids, carbohydrates, proteins, and nucleic acids [209], giving bacteria a 3D protective structure. Bacteria implanted in the matrix can interact synergistically, communicate with one another, and transfer resistance genes [214]. Additionally, the lower oxygen and nutrition levels in the higher depth of the matrix lead to the creation of latent persister cells, which support antimicrobial tolerance and resistance [212,215]. The matrix is also rich in hydrophobic groups and negatively charged elements and 3D structure of EPS forming mesh-like structure containing pores filled with water, facilitating transmission between bacterial communities [216,217].

MDR biofilm infections represent an especially challenging treatment obstacle, and to combat biofilms, it is necessary to overcome the physical barrier produced by them [215,218,219].

Bacterial communities communicate with each other through autoinducer called quorum sensing (QS); therefore, biofilm eradication occurs through degrading EPS, QS, and killing of internal bacteria [220].

Nanoparticles show promising effect to overcome and eradicate MDR biofilm infections. The invasion of this highly stable polymeric matrix can be enhanced by optimizing the surface properties and designating the nanoparticles [25,221]. Important variables affecting the invasion and penetration of biofilm include size and electrostatic interactions. Generally, positively charged nanomaterials have an excellent penetration within the matrix, whereas uncharged nanomaterials smaller than 350 nm are more mobile through holes present in biofilms [222].

When nanomaterials invade biofilms, they can interact with bacteria and produce antimicrobial effects. It wase proved in previous studies that polymeric nano delivery systems, in which the antibiotic is encased in a formulation, have shown considerable promise in terms of facilitating a high local drug concentration at the site of infection, controlled drug release, and minimal drug degradation [223]. In another study, zinc oxide nanoparticles with pancreatin enzyme (ZnONPs-PK) have anti-bacterial and anti-biofilm activity and are potent to eradicate MRSA [224]. In a different method, a promising result was achieved when using prolonged release poly-lactic-co-glycolic acid (PLGA) micro- and nanoparticles containing ciprofloxacin against biofilms of *S. aureus* and *P. aeruginosa* [225].

## 8. Strengths and Challenges in the Application of NPs against MDR Infection

Due to the unique size and physical characters of NPs, they provide significant advantages over conventional antibiotics; however, their delivery method is still challenging. They provide sustained release system and targeted delivery of antibiotic. Therefore, small and reduced doses are the required result in order to decrease the frequency of dosing, host cell toxicity and emergence of resistance [226,227].

However, NPs systemic administration still has to handle a number of other issues and challenges, including additional analysis of NP interactions with cells, tissues, and organs, optimal dose, identification of suitable routes of administration, toxicity following both short-term and long-term exposure [227,228], and exact mechanism of cellular uptake. Moreover, prolonged exposure to NPs in the workplace may result in unanticipated health risks. Inhaling nanoparticles in the form of air pollutants is another method of secondary exposure to NPs. Sometimes, these inhaled nanoparticles can evade the immune system and spread throughout the body, causing systemic health problems. Moreover, intravenous administration of NPs lead to their accumulation in spleen, lung and bone marrow [229]. In addition, inhaled NPs can distribute through lung, heart, liver, spleen and brain due to their small size, large surface area and high absorption rate [230]. Administration of NP therapeutically may cause multiorgan nanotoxicity. In all toxic cases, NPs are linked to oxidative stress, which in turn causes hepatotoxicity and lung toxicity [231] and metabolic modifications such as β oxidation of fatty acid, glycolysis and reduced ketogenesis that occur through ROS and may be associated with hepatotoxicity and nephrotoxicity [232]. Therefore, due to previous challenges, no FDA-approved nAbts for systemic human use have been launched to date. Hence, the safety, toxicity, pharmacokinetic and pharmacodynamic profile of NP systems as well as the therapeutic efficacy measures used in clinical trials must be considered in future preclinical investigations [11,22,226,233]. Finally, the financial implications of these NP clinical translations regarding their therapeutic efficacy need to be evaluated [226,234].

## 9. Conclusions and Future Perspectives

The emergence and growth of MDR bacteria have shown to be a serious health issue that must be managed on a global scale. To treat persistent MDR planktonic bacterial and biofilm infections, nanomaterials offer a novel, “outside the box” strategy. Optimization of their physical characters, particularly size and surface charge, is essential to maximize their curative potential and decrease host hazard. Clinical use is currently limited by issues concerning the long-term effects of nanoparticles on the person and systemic safety. Moreover, metabolic, toxicity, stability issues and mechanism explanation on the gene level show that the NP effect on gene expression should be explained in detail in future studies. Additionally, the exact and detailed mechanism of interaction of NPs with biological systems should be explained in future trials to develop nanomaterials with favorable physicochemical properties that will allow them to be more responsive to varied biological settings for therapeutic advantages while having no negative effect. The field of nAbs will act as the next-generation medicines in near future to combat MDR bacteria.

## Figures and Tables

**Figure 1 biomedicines-11-00413-f001:**
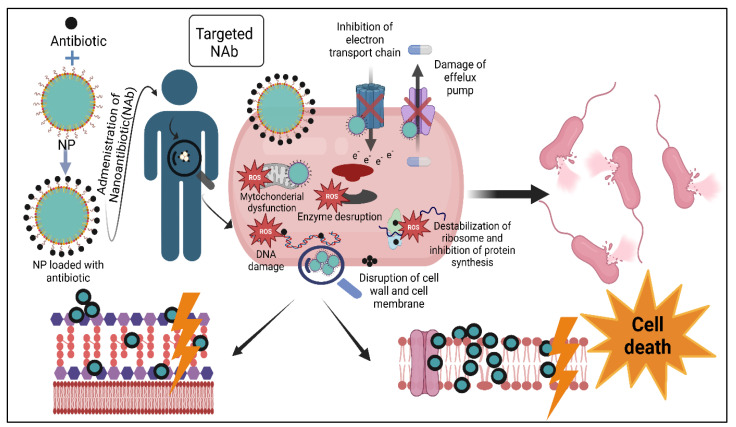
Loading NPs with conventional antibiotics. NPs act as a delivery system to transfer Ab to its target in resistant microorganism. This provides dual mechanisms to combat MDR organism through the effect of Ab and the effect of NPs itself with net result potentiation of bacterial cell death. Abs exert their bactericide effect through disruption of bacterial cell wall or cell membrane, destabilization of ribosomes and inhibition of protein synthesis or inhibition of nucleic acid synthesis. NPs may act through more than one mechanism; disruption of cell wall and cell membrane, mitochondrial damage, enzyme damage, protein damage, efflux pump damage, inhibition of electron transport chain and production of ROS that generate oxidative stress and lead to damage of vital cellular macromolecules.

**Figure 2 biomedicines-11-00413-f002:**
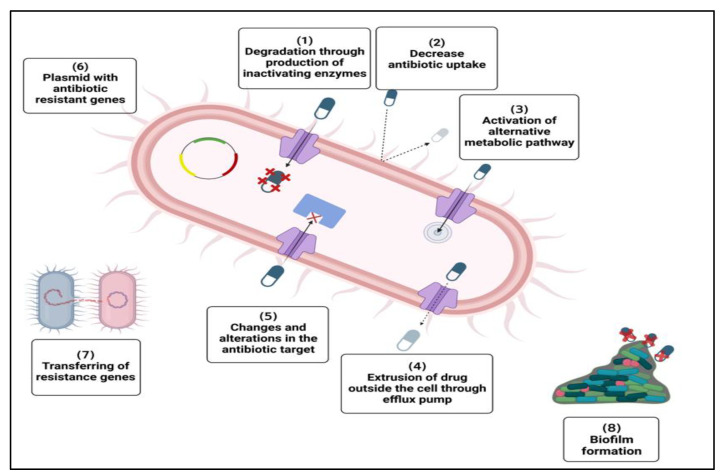
Mechanisms of development of antibiotic resistance. It develops through different mechanisms: (1) breakdown of drugs through production of inactivating enzyme, (2) decrease in antibiotic uptake, (3) alteration of common metabolic pathway, (4) extrusion of drug outside the cell by efflux pump, (5) change and alteration in antibiotic target, (6) presence of plasmid carrying multiple resistance genes, (7) sharing resistance gene between different bacteria, (8) formation of protective extracellular polymeric matrix.

**Figure 3 biomedicines-11-00413-f003:**
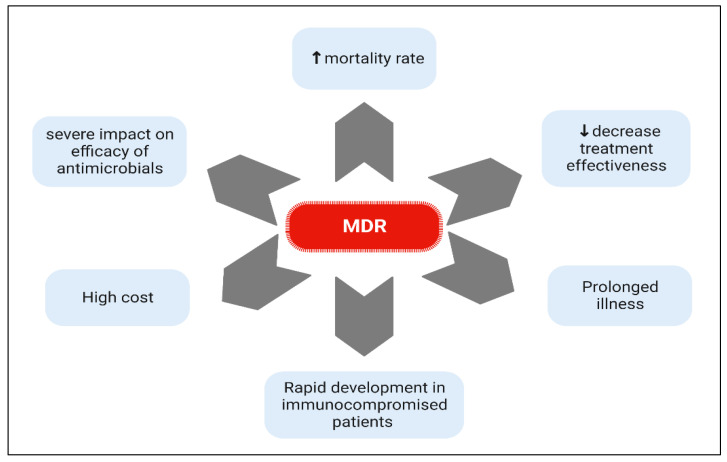
Risk factors related to MDR. MDR increase the risk of mortality rate, decrease in medication efficacy, prolong the period of illness, rapidly develop in immunocompromised patients, are not affected by most antimicrobials and need more expensive alternatives.

**Figure 4 biomedicines-11-00413-f004:**
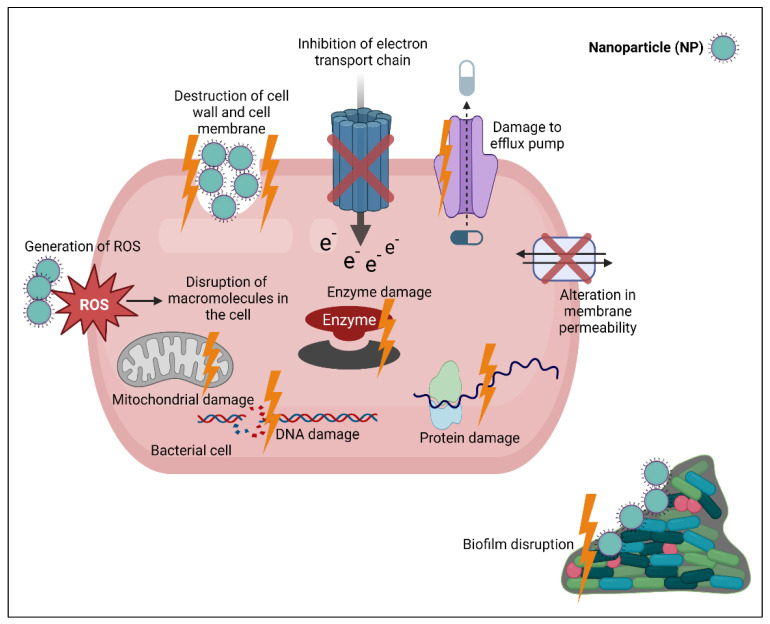
Mechanisms of action of NPs. They act through destruction of bacterial cell wall and cell membrane, overproduction of reactive oxygen species (ROS) that produce oxidative stress and damage vital intracellular macromolecules, as well as inflict protein damage, enzyme damage, mitochondrial damage, DNA damage, alteration in membrane permeability, prevent extrusion of drug through damage of efflux pump, inhibition of electron transport chain and prevent and disrupt biofilm.

**Figure 5 biomedicines-11-00413-f005:**
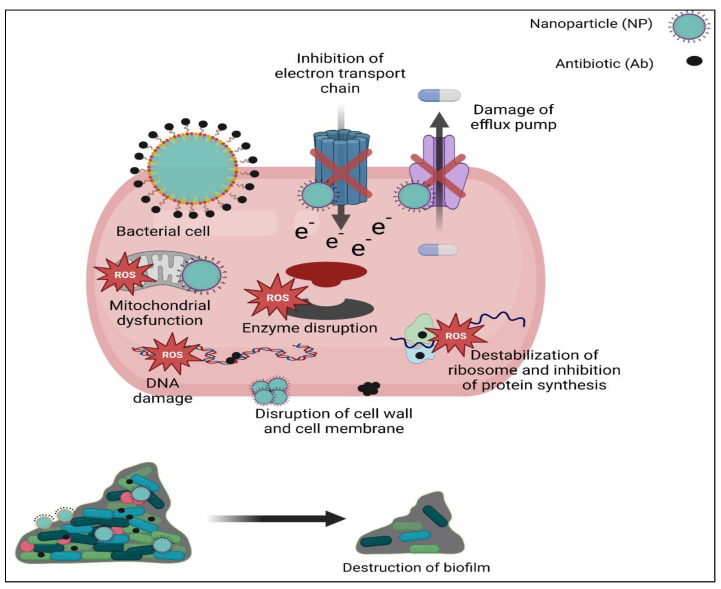
Synergistic activity of nanoparticles loaded with antibiotics. Combined nanoantibiotics (nAbs) exert dual antibacterial action through the effect of nanoparticles and antibiotics. Antibiotics can exert rapid and potent action through targeted delivery by NPs and exert their bactericidal effect through disruption of bacterial cell wall or cell membrane, destabilization of ribosomes and inhibition of protein synthesis or inhibition of nucleic acid synthesis. In addition, NPs can cause damage to bacterial cell wall, cell membrane, proteins, DNA, enzymes, mitochondria, efflux pump and inhibit electron transport and release reactive oxygen species (ROS) that cause damage to vital cellular macromolecules.

## Data Availability

Not applicable.

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
