# Peer review of "Nanotechnology as a Promising Approach to Combat Multidrug Resistant Bacteria: A Comprehensive Review and Future Perspectives"

_biomedicines, 2023, doi:10.3390/biomedicines11020413_

Round 1

Reviewer 1 Report

Overall the review article is on. However, the author needs to add more information in some of the sections of the article, especially the mechanism of action,  Strengths and challenges on the application of NP against MDR infection, and the Conclusions and future perspectives. The author should discuss the mechanism of action in detail with more figures and illustrations. Besides, the conclusion and future prospects sections are too small, the author should elaborate on them in detail. There are only 2 figures and a graphical abstract in the manuscript, the author should add more tables and figures in the manuscript in order to make it interesting.

Besides, there are already a number of publisher articles related to the current topic, the author should explain how their manuscript is different from previously published articles and what is the novelty of their article.

1. Nanotechnology as a Novel Approach in Combating Microbes Providing an Alternative to Antibiotics. Antibiotics 202110(12), 1473; https://doi.org/10.3390/antibiotics10121473

2. Overcoming Multidrug Resistance in Bacteria Through Antibiotics Delivery in Surface-Engineered Nano-Cargos: Recent Developments for Future Nano-Antibiotics. Front. Bioeng. Biotechnol., 08 July 2021.

Author Response

Dear Rev 1
We have attached word file with the answers.

Thanks for the valuable suggestions

Kind regards

Reviewer 2 Report

Helal et al. present Nanotechnology as a promising approach to combat multidrug-resistant bacteria: A comprehensive review and future perspectives. This review described here is to regard the approach to the current advances in nanotechnology to combat MDR and biofilm infection. 

In my opinion, this work is of interest to researchers in the field of a promising approach to combat multidrug-resistant bacteria. But it requires major revision before it becomes suitable for publication. The authors should consider the following comments to improve their manuscript. And, the English needs to be reviewed by a native English speaker.

MA1. Graphical abstract should be replaced to Figure 1. And it should be described in more details.

MA2. The Introduction must be improved by incorporating more recent references including 

nanotechnology as a promising approach to combat multidrug-resistant bacteria.

MA3. Figures 1 and 2 should be described in more detail in the manuscript.

MA4.  Please address more details of synergistic activity of nanoantibiotics.

MA5. In conclusion, please the contents detailed should be addressed including future scope and applications for better understanding of the strength and limitation of the application of nanotechnology against bacterial infection and future perspectives.

The subject may be interesting enough biomedicines but only after major, deep revision, if at all possible, to resolve the above.

Author Response

Dear Rev 2
We have attached word file with the answers.

Thanks for the valuable suggestions

Kind regards

Round 2

Reviewer 1 Report

The revision is ok.

Reviewer 2 Report

The authors have improved the manuscript for publication.